# Semantic-Guided Multi-Attention Localization for Zero-Shot Learning

**Yizhe Zhu**[*]
Rutgers University
yizhe.zhu@rutgers.edu,

**Jianwen Xie**
Hikvision Research Institute
jianwen@ucla.edu

**Zhiqiang Tang**
Rutgers University
zhiqiang.tang@rutgers.edu,

**Xi Peng**
University of Delaware
xipeng@udel.edu

**Ahmed Elgammal**
Rutgers University
elgammal@cs.rutgers.edu

## Abstract

Zero-shot learning extends the conventional object classification to the unseen class recognition by introducing semantic representations of classes. Existing approaches predominantly focus on learning the proper mapping function for visual-semantic embedding, while neglecting the effect of learning discriminative visual features. In this paper, we study the significance of the discriminative region localization. We propose a semantic-guided multi-attention localization model, which automatically discovers the most discriminative parts of objects for zero-shot learning without any human annotations. Our model jointly learns cooperative global and local features from the whole object as well as the detected parts to categorize objects based on semantic descriptions. Moreover, with the joint supervision of embedding softmax loss and class-center triplet loss, the model is encouraged to learn features with high inter-class dispersion and intra-class compactness. Through comprehensive experiments on three widely used zero-shot learning benchmarks, we show the efficacy of the multi-attention localization and our proposed approach improves the state-of-the-art results by a considerable margin.

## 1 Introduction

Deep convolutional neural networks have achieved significant advances in object recognition. The main shortcoming of deep learning methods is the inevitable requirement of large-scale labeled training data that needs to be collected and annotated by costly human labor [1, 2, 3, 4]. In spite that images of ordinary objects can be readily found, there remains a tremendous number of objects with insufficient and scarce visual data [5]. This attracts many researchers' interest in how to recognize objects with few or even no training samples, which are known as few-shot learning [6, 7] and zero-shot learning [8, 9, 5, 10, 11, 12], respectively.

Zero-shot learning mimics the human ability to recognize objects only from a description in terms of concepts in some semantic vocabulary [13]. The underlying key is to learn the association between visual representations and semantic concepts and use it to extend the possibility to unseen object recognition. In a general sense, the typical scheme of the state-of-the-art approaches of zero-shot learning is (1) to extract the feature representation of visual data from CNN models pretrained on the large-scale dataset(e.g., ImageNet), (2) to learn mapping functions to project the visual features and semantic representations to shared space. The mapping functions are optimized by either ridge regression loss [14, 15] or ranking loss on compatibility scores of two mapped features [8, 9]. Taking

---

[*]Work was done while Yizhe Zhu was an intern at Hikvision Research Institute.

advantage of the success of generative models (e.g., GAN [16], VAE [17], generator network [18]) in data generation, several recent methods [5, 10, 11] resort to hallucinating visual features of unseen classes, converting zero-shot learning to conventional object recognition problems.

All aforementioned methods neglect the significance of discriminative visual feature learning. Since the CNN models are pretrained on a traditional object recognition task, the extracted features may not be representative enough for zero-shot learning task. Especially in the fine-grained scenarios, the features learned from the coarse object recognition can hardly capture the subtle difference between classes. Although several recent works [13, 19] solve the problem in an end-to-end manner that is capable of discovering more distinctive visual information suitable for zero-shot recognition, they still simply extract the global visual feature of the whole image, without considering the effect of discriminative part regions in the images. We argue that there are multiple discriminative part areas that are key points to recognize objects, especially fine-grained objects. For instance, the head and tail are crucial to distinguish bird species. To capture such discriminating regions, we propose a semantic-guided attention localization model to pinpoint, where the most significant parts are. The compactness and diversity loss on multi-attention maps are proposed to encourage attention maps to be compact in the most crucial region in each map while divergent across different attention maps.

We combine the whole image and multiple discovered regions to provide a richer visual expression and learn global and local visual features (i.e., image features and region features) for the visual-semantic embedding model, which is trained in an end-to-end fashion. In the zero-shot learning scenario, embedding softmax loss [13, 19] is used by embedding the class semantic representations into a multi-class classification framework. However, softmax loss only encourages the inter-class separability of features. The resulting features are not sufficient for recognition tasks [20]. To encourage high intra-class compactness, class-center triplet loss [21] assigns an adaptive "center" for each class and forces the learned features to be closer to the "center" of the corresponding class than other classes. In this paper, we involve both embedding softmax loss and class-center triplet loss as the supervision of feature learning. We argue that these cooperative losses can efficiently enhance the discriminative power of the learned features.

To the best of our knowledge, this is the first work to jointly optimize multi-attention localization with global and local feature learning for zero-shot learning tasks in an end-to-end fashion. Our main contributions are summarized as follows:

- We present a weakly-supervised multi-attention localization model for zero-shot recognition, which jointly discovers the crucial regions and learns feature representation under the guidance of semantic descriptions.

- We propose a multi-attention loss to encourage compact and diverse attention distribution by applying geometric constraints over attention maps.

- We jointly learn global and local features under the supervision of embedding softmax loss and class-center triplet loss to provide an enhanced visual representation for zero-shot recognition.

- We conduct extensive experiments and analysis on three zero-shot learning datasets and demonstrate the excellent performance of our proposed method on both part detection and zero-shot learning.

## 2 Related Work

**Zero-Shot Learning Methods** While several early works of zero-shot learning [22] make use of the attribute as the intermediate information to infer the labels of images, the current majority of zero-shot learning approaches treat the problem as a visual-semantic embedding one. A bilinear compatibility function between the image space and the attribute space is learned using the ranking loss in ALE [8] or the ridge regression loss in ESZSL [23]. Some other zero-shot learning approaches learn non-linear multi-model embeddings; for example, LatEm [9] learns a piecewise linear model by the selection of learned multiple linear mappings. DEM [14] presents a deep zero-shot learning model with non-linear activation ReLU. More related to our work, several end-to-end learning methods are proposed to address the pitfall that discriminative feature learning is neglected. SCoRe [13] combines two semantic constraints to supervise attribute prediction and visual-semantic embedding, respectively. LDF [19] takes one step further and integrates a zoom network in the model to discover

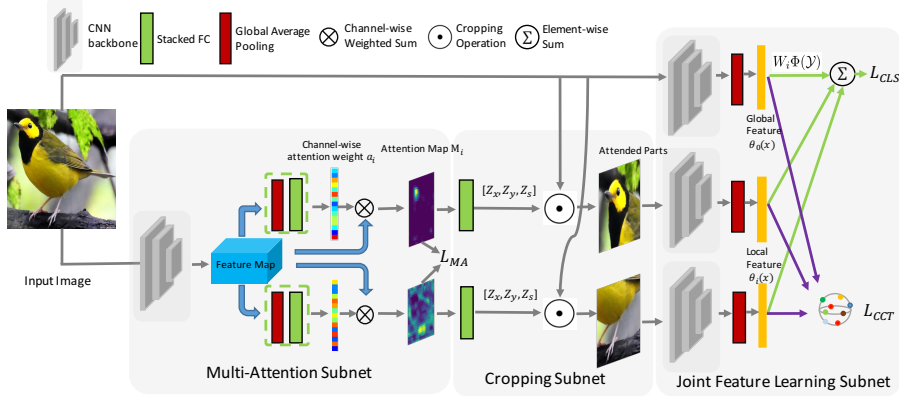

Figure 1: The Framework of the proposed Semantic-Guided Multi-Attention localization model (SGMA). The model takes as input the original image and produces $n$ part attention maps (here $n = 2$). Multi-Attention loss $\mathcal{L}_{MA}$ keeps the attention areas compact in each map and divergent across different maps. The part images from the cropping subnet and the original images are fed into different CNNs in the joint feature learning subnet for semantic description-guided object recognition.

significant regions automatically, and learn discriminative visual feature representation. However, the zoom mechanism can only discover the whole object by cropping out the background with a square shape, still being restricted to the global features. In contrast, our multi-attention localization network can help find multiple finer part regions (e.g., head, tail) that are discriminative for zero-shot learning.

**Multi-Attention Localization** Several previous methods are proposed to leverage the extra annotations of part bounding boxes to localize significant regions for fine-grained zero-shot recognition [24, 25, 5]. [24] straightforwardly extracts the part features by feeding annotated part regions into a CNN pretrained on ImageNet dataset. [25, 5, 26] train a multiple-part detector with groundtruth annotations to produce the bounding boxes of parts and learn the part features with conventional recognition tasks. However, the heavy involvement of human labor for part annotations makes tasks costly in the real large-scale problems. Therefore, learning part attentions in a weakly supervised way is desirable in the zero-shot recognition scenario. Recently, several attention localization models are presented in the fine-grained classification scenario. [27, 28] learn a set of part detectors by analyzing filters that consistently respond to specific patterns. Spatial transformer [29] proposes a learnable module that explicitly allows the spatial manipulation of data within the network. In [30, 28, 31], candidate part models are learned from convolutional channel responses. Our work is different in three aspects: (1) we learn part attention models from convolutional channel responses; (2) instead of using the supervision of the classification loss, our model discovers the parts with semantic guidance, making the located part more discriminative for zero-shot learning; (3) zero-shot recognition model and attention localization model are trained jointly to ensure the parts localization are optimized for the zero-shot object recognition.

## 3 Method

We start by introducing some notations and the problem definition. Assume there are $N$ labeled instances from $C^s$ seen classes $\mathcal{D}^s = \{(x_i^s, y_i^s, s_i^s)\}_{i=1}^N$ as training data, where $x_i^s \in \mathcal{X}$ denotes the image, $y_i^s \in \mathcal{Y}^s$ is the corresponding class label , $s_i^s = \phi(y_i^s) \in \mathcal{S}$ is the semantic representation of the corresponding class. Given an image $x_i^u$ from an unseen class and a set of semantic representations of unseen classes $\{s_i^u = \phi(y_i^u)\}_{i=1}^{C^u}$, where $C^u$ denotes the number of unseen classes, the task of zero-shot learning is to predict the class label $y^u \in \mathcal{Y}^u$ of the image, where $\mathcal{Y}^s$ and $\mathcal{Y}^u$ are disjoint.

The framework of our approach is demonstrated in Figure 1. It consists of three modules: the multi-attention subnet, the region cropping subnet, the joint feature embedding subnet. The multi-attention subnet generates multiple attention maps corresponding to distinct parts of the object. The region cropping subset crops the discriminitive parts with differentiable operations. The joint feature learning subnet takes as input the cropped parts and the original image, and learns the global and local visual feature for the final zero-shot recognition.

### 3.1 Multi-Attention Subnet

LDF [19] presents a cascaded zooming mechanism to localize the object-centric region gradually while cropping out background noise. Different from LDF, our method considers multiple finer discriminative areas, which can provide various richer cues for object recognition. Our approach starts with the multi-attention subnet to produce attention maps.

As shown in Figure 1, the input images first pass through the convolutional network backbone to become feature representations of size $H \times W \times C$. The attention weight vectors $a_i$ over channels are obtained for each attended part based on the extracted feature maps. The attention maps are finally produced by the weighted sum of feature maps over channels with the previously obtained attention weight vectors $a_i$. To encourage different attention maps to discover different discriminating regions, we design compactness and diversity loss. Details will be shown later.

To be specific, the channel descriptor $p$, encoding the global spatial information, is first obtained by using global average pooling on the extracted feature maps. Formally, the features are shrunk through its spatial dimension $H \times W$. The $c^{th}$ element of $p$ is calculated by:

$$p_c = \frac{1}{H \times W} \sum_{i=1}^{H} \sum_{j=1}^{W} b_c(i,j), \tag{1}$$

where $b_c$ is the feature in $c^{th}$ channel. To make use of the information of the channel descriptor $p$, we follow it by the stacked fully connected layers to fully capture channel-wise dependencies of each part. A sigmoid activation $\sigma(\cdot)$ is then employed to serve as a gating mechanism. Formally, the channel-wise attention weight $a_i$ is obtained by

$$a_i = \sigma(W_2 f(W_1 p)), \tag{2}$$

where $f(\cdot)$ refers to the ReLU activation function and $a_i$ can be considered as the soft-attention weight of channels associated with the $i^{th}$ part. As discovered in [32, 33], each channel of features focuses on a certain pattern or a certain part of the object. Ideally, our model aims to assign high weights (i.e., $a_i^c$) to channels associated with a certain part $i$, while giving low weights to channels irrelevant to that part.

The attention map for $i^{th}$ part is then generated by the weighted sum of all channels followed by the sigmoid activation:

$$M_i(x) = \sigma(\sum_{c=1}^{C} a_i^c f_{Conv}(x)^c), \tag{3}$$

where the superscript $c$ means $c^{th}$ channel, and $C$ is the number of channels. For brevity, we omit $(x)$ in the rest of the paper. With the sigmoid activation, the attention map plays the role of gating mechanism as in the soft-attention scheme, which will force the network to focus on the discriminative parts.

**Multi-Attention Loss**

To enable our model to discover diverse regions over attention maps, we design the multi-attention loss by applying the geometric constraints. The proposed loss consists of two components:

$$\mathcal{L}_{MA} = \sum_{i}^{N_a} [\mathcal{L}_{CPT}(M_i) + \lambda \mathcal{L}_{DIV}(M_i)], \tag{4}$$

where $\mathcal{L}_{CPT}(M_i)$ and $\mathcal{L}_{DIV}(M_i)$ are compactness and diversity losses respectively, $\lambda$ is a balance factor, and $N_a$ is the of attention maps. Ideally, we want the attention map to concentrate around the peak position rather than disperse. The ideal concentrated attention map for the part $i$ is created as a Gaussian blob with the Gaussian peak at the peak activation of the $i^{th}$ attention map. Let $z$ be the position of the attention map, and $\mathcal{Z}$ be the set of all positions. The compactness loss is defined as follows:

$$\mathcal{L}_{CPT}(M_i) = \frac{1}{|\mathcal{Z}|} \sum_{z \in \mathcal{Z}} ||m_i^z - \widetilde{m}_i^z||_2^2, \tag{5}$$

where $m_i^z$ and $\widetilde{m}_i^z$ denote the generated attention map and the ideal concentrated attention map at location $z$ for $i^{th}$ part respectively, and $|\mathcal{Z}|$ denotes the size of attention maps. The $L_2$ heatmap

regression loss has been widely used in human pose estimation scenarios to localize the keypoints [34, 35], but here we use it for a different purpose.

Intuitively, we also expect the attention maps to attend different discriminative parts. For example, one map attends the head while another map attends the tail. To fulfill this goal, we design a diversity loss $\mathcal{L}_{DIV}$ to encourage the divergent attention distribution across different attention maps. Formally, it is formulated by:

$$\mathcal{L}_{DIV}(M_i) = \sum_{z \in \mathcal{Z}} m_i^z \max\{0, \widehat{m}^z - mrg\}, \tag{6}$$

where $\widehat{m}^z = \max_{k \neq i} m_k^z$ represents the maximum of other attention maps at location $z$ and $mrg$ denotes a margin. The maximum-margin design here is to make the loss less sensitive to noises and improve the robustness. The motivation of the diversity loss is that when the activation of a particular position in one attention map is high, the loss prefers lower activations of other attention maps in the same position. From another perspective, $\mathcal{L}_{DIV}$ can be roughly considered as the inner product of two flattened matrices, which measures the similarity of two attention maps.

## 3.2 Region Cropping Subnet

With the attention maps in hand, the region can be directly cropped with a square centered at the peak value of each attention map. However, it's hard to optimize such a non-continuous cropping operation with backward propagation. Similar to [36, 19] , we design a cropping network to approximate region cropping. Specifically, with an assumption of a square shape of the part region for computational efficiency, our cropping network takes as input the attention maps from the multi-attention subnet, and outputs three parameters:

$$[t_x, t_y, t_s] = f_{CNet}(M_i), \tag{7}$$

where $f_{CNet}(\cdot)$ denotes the cropping network and consists of two FC layers, $t_x$, $t_y$ represent the x-axis and y-axis coordinates of the square center respectively, $t_s$ is the side length of the square. We produce a two-dim continuous boxcar mask $V(x, y) = V_x \cdot V_y$:

$$\begin{aligned} V_x &= f(x - t_x + 0.5t_s) - f(x - t_x - 0.5t_s), \\ V_y &= f(y - t_y + 0.5t_s) - f(y - t_y - 0.5t_s), \end{aligned} \tag{8}$$

where $f(x) = 1/(1 + \exp(-kx))$. The cropped region is obtained by the element-wise multiplication between the original image and the continuous mask, $x_i^{part} = x \odot V_i$, where $i$ is the index of parts. We further utilize the bilinear interpolation to resize the cropped region $x_i^{part}$ to the same size of the original images. Interested readers are referred to reference [36] for details.

## 3.3 Joint Feature Learning Subnet

To provide enhanced visual representations of images for zero-shot learning, we jointly learn the global and local visual features given the original image and part images produced by the region cropping subnet. As shown in Figure 1, the original image and part patches are resized to $224 \times 224$ and fed into separate CNN backbone networks (with the identical VGG19 architecture). The convolution layers are followed by the global average pooling to get the visual feature vector $\theta(x)$.

To learn the discriminative features for the zero-shot learning task, we employ two cooperative losses: the embedding softmax loss $\mathcal{L}_{CLS}$ and the class-center triplet loss $\mathcal{L}_{CCT}$. The former encourages a higher inter-class distinction, while the latter forces the learned feature of each class to be concentrated with a lower intra-class divergence.

**Embedding Softmax Loss**

Let $\phi(y)$ denote the semantic feature. The compatibility score of multi-model features is defined as $s = \theta(x)^T W \phi(y)$, where $W$ is a trainable transform matrix. If the compatibility scores are considered as logits in softmax, the embedding softmax loss can be given by:

$$\mathcal{L}_{CLS} = -\frac{1}{N} \log \frac{\exp(s_j)}{\sum_{\mathcal{Y}_s} \exp(s_j)}, \tag{9}$$

where $s_j = \theta(x)^T W \phi(y_j)$, $y_j \in \mathcal{Y}_s$, and $N$ is the number of training samples. In order to combine the global and local features without increasing the complexity of the model, we adopt the late fusion

strategy. The overall compatibility scores are obtained by summing up the compatibility scores from each CNN and used to compute the softmax loss. Note that the strategy can significantly reduce the number of parameters of the network by discarding the additional dimension reduction layer (i.e., FC layer) after the feature concatenation used in [19]. Formally, we substitute $s_j$ in Eq. 9 with $\sum_i (s_j^i)$, where $s_j^i = \theta_i(x)^T W_i \phi(y_j)$ and $i$ is the index of part images and the original image.

**Class-Center Triplet Loss**

The class-center triplet loss [21] is originally designed to minimize the intra-class distances of deep visual features in face recognition tasks. In our case, we jointly train the network with the class-center triplet loss to encourage the intra-class compactness of features. Let $i, k$ be the class indices, the loss is formulated as:

$$\mathcal{L}_{CCT} = \max\{0, mrg + ||\widehat{\phi}_i - \widehat{C}_i||_2^2 - ||\widehat{\phi}_i - \widehat{C}_k||_2^2\}_{i \neq k}, \tag{10}$$

where $mrg$ is the margin, $\phi_i$ is the mapped visual feature in semantic feature space (i.e., $\phi_i = \theta(x)^T W_i$), $C_i$ denotes the "center" of each class that are trainable parameters, $\widehat{\cdot}$ means $L_2$ normalization operation. The normalization operation is involved to make feature points located on the surface of a unit hypersphere, leading to the ease of setting the proper margin. Moreover, class-center triplet loss exempts the necessity of triple sampling in the naive triplet loss.

Overall, the proposed SGMA model is trained in an end-to-end manner with the objective:

$$\mathcal{L}_{SGMA} = \mathcal{L}_{MA} + \alpha_1 \mathcal{L}_{CLS} + \alpha_2 \mathcal{L}_{CCT}, \tag{11}$$

where the balance factor $\alpha_1$ and $\alpha_2$ are consistently set to 1 in all experiments.

## 3.4 Inference from SGMA Model

We provide two ways to infer the labels of unseen class images from the SGMA model. The first one is straightforwardly to choose the class label with the maximal overall compatibility score, as the green path in Figure 1. An alternative way is utilizing the features $\phi_{cct}(x)$ learned in the class-center branch, as the purple path in Figure 1. The class label can be inferred based on the similarities between the feature of the test image $\phi_{cct}(x)$ and the prototypes of unseen classes $\Phi_{cct}^u$, which can be obtained by the following steps. We assume the semantic descriptions of unseen classes can be represented by a linear combination of those of seen classes. Let $W$ be the weight matrix of such a combination, and $W$ can be obtained by solving the ridge regression:

$$W = \arg\min_W ||\Phi^u - W\Phi^s||_2^2 + \lambda||W||_2^2, \tag{12}$$

where $\Phi^u$ and $\Phi^s$ are the semantic matrices of unseen and seen classes with each row being the semantic vector of each class. Equipped with the learned $W$ describing the relationship of the semantic vectors of seen and unseen classes, we can obtain the prototypes for unseen classes by applying the same $W$, $\Phi_{cct}^u = W\Phi_{cct}^s$, where $\Phi_{cct}^s$ is the prototypes of seen classes obtained by averaging the features of all images of each class.

To combine the global and local descriptions of images, we concatenate the visual features generated by different CNNs. Moreover, to combine the inference of two ways, the compatibility scores from the embedding softmax branch and the similarity scores from the class-agent triplet branch are added as the final prediction scores of the test image w.r.t. unseen classes:

$$y = \arg\min_{y \in \mathcal{Y}_u} (s_y + \beta \langle \phi_{cct}(x), [\Phi_{cct}^u]_y \rangle), \tag{13}$$

where $s_y = \theta(x)^T W \phi(y)$, $\langle \cdot \rangle$ denotes inner product, $[\cdot]_y$ denotes the row of the matrix corresponding to the class $y$, and $\beta$ a balancing factor to control the contribution of the class-center branch.

# 4 Experiment

To evaluate the empirical performance of our proposed approach, we conduct experiments on three standard zero-shot learning datasets and compare our method with the state-of-the-art ones. We then show the performance of multi-attention localization. In our experiment, we only use two attention maps as we find that more maps will cause severe overlap among attended regions and hardly improve the zero-shot learning performance.

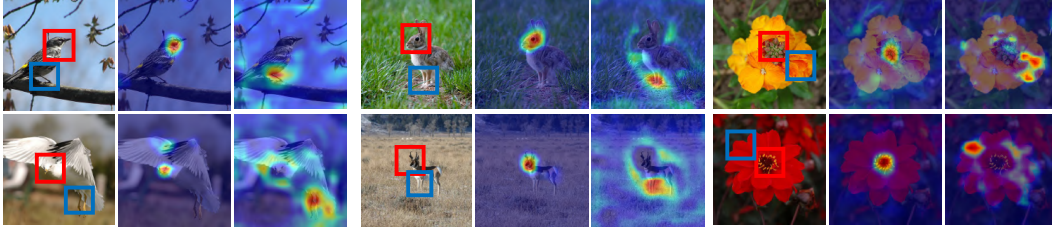

Figure 2: Part detection results on three benchmarks. Each row displays three examples of results. Each result consists of three images, where the detected parts are marked with blue and red bounding boxes in the first image, and the rest two images are the corresponding generated attention maps.

## 4.1 Implementation Details and Model Initialization

We implement our approach on the Pytorch Framework. For the multi-attention subnet, we take the images of size $448 \times 448$ as input in order to achieve high-resolution attention maps. For the joint feature embedding subnet, we resize all the input images to the size of $224 \times 224$. We consistently adopt VGG19 as the backbone and train the model with a batch size of 32 on two GPUs (TitanX). We use the SGD optimizer with the learning rate of $0.05$, the momentum of $0.9$, and weight decay of $5 * 10^{-4}$ to optimize the objective functions. The learning rate is decay by $0.1$ on the plateau, and the minimum one is set to be $5 * 10^{-4}$. Hyper-parameters in our models are obtained by grid search on the validation set. $mrg$s in Eq. 7 and Eq. 10 are set to be $0.2$ and $0.8$, respectively. $k$ in Eq. 8 is set to be 10. The number of parts is set to be 2 since we find that increasing the number of parts will result in little improvement on the zero-shot learning performance and lead to attention redundancy, i.e., maps attend to the same region.

For multi-attention subnet, we apply unsupervised k-means clustering to group channels based on the peak activation positions and initialize $a_i$ with the pseudo labels generated by the clustering. Interested readers are referred to reference [36] for details. The attention maps from the initialized multi-attention subnet are leveraged to pretrain the region cropping subnet. Specifically, we obtain the attended region in attention maps by a discriminative square centered at the peak response of the attention map ($[p_x, p_y]$). The side length of the squares $t_s$ is assumed to be the quarter of the image size. The coordinates of the attended region ($[p_x, p_y, t_s]$) are considered as pseudo ground truths to pretrain the cropping subnet with MSE loss, and the attended regions are utilized as the cropped parts to pretrain the joint feature learning subnet.

## 4.2 Datasets and Experiment Settings

We use three widely used zero-shot learning datasets: Caltech-UCSD-Birds 200-2011 (CUB) [37], Oxford Flowers (FLO) [38], Animals with Attributes (AwA) [22]. CUB is a fine-grained dataset of bird species, containing 11,788 images from 200 different species and 312 attributes. FLO is a fine-grained dataset, consisting of 8,189 images from 102 different types of flowers without attribute annotations. However, the visual descriptions are available and collected by [39]. Finally, AwA is a coarse-grained dataset with 30,475 images, 50 classes of animals, and 85 attributes.

To fairly compare with baselines, we use the attributes or sentence features provided by [40, 10] as semantic features for all methods. For non-end-to-end methods, we consistently use 2,048-dimensional features extracted from a pretrained 101-layer ResNet provided by [40], and for end-to-end methods, we adopt VGG19 as the backbone network. Besides, [40] points out that several test classes in the standard splitting (marked as SS) of zero-shot learning setting are utilized for training the feature extraction network, which violates the spirit of zero-shot that test classes should never be seen before. Therefore, we also evaluate methods on the splitting proposed by [10] (marked as PS). We measure the quantitative performance of the methods in terms of Mean Class Accuracy (MCA).

## 4.3 Part Detection Results

To evaluate the efficacy of weakly supervised part detection, we compare our detection results on CUB with SPDA-CNN [41], a state-of-the-art work on part detectors trained with ground truth part annotations. We observe our model consistently attend the head or tail on two attention maps

respectively. Therefore, we compare the detected parts with head or tail ground truth annotations. Part detection is considered correct if it has at least 0.5 overlap with ground truth (i.e., IoU $> 0.5$).

As shown in Table 1, the SPDA-CNN can be considered the upper bound since it leverages part annotation to train detectors. We also provide the results of random crops that serve as a lower bound. Compared with the random crops, our method has achieved an improvement of 35.7% on average. Although there is still a small gap between the performances of ours and SPDA-CNN (61.5%v.s.79.1%) due to the lack of precise part annotations, the results are promising since our model is more practical in the large-scale real-world tasks where costly annotations are not available. Besides, if we remove the proposed multi-attention loss (marked as "ours w/o MA"), the performance suffers a significant drop (47.6% v.s. 61.5%), confirming the effect of the multi-attention loss.

Table 1: Part detection results measured by average precision(%).

| Method | Head | Tail | Average |
|---|---|---|---|
| SPDA-CNN | 90.9 | 67.2 | 79.1 |
| Ours | 74.9 | 48.1 | 61.5 |
| Ours w/o MA | 65.7 | 29.4 | 47.6 |
| Random | 25.6 | 26.0 | 25.8 |

We also show the qualitative results of part localization in Figure 2. The detected parts are well-aligned with semantic parts of objects. In CUB, two parts are associated with the head and the legs of birds, while the parts are the head and rear body of the animals in AwA. In FLO, the stamen and pistil are roughly detected in the red box, while the petal is localized as another crucial part.

Table 2: Zero-shot learning results on CUB, AWA, FLO benchmarks. The best scores and second best ones are marked bold and underline respectively.

| Method | CUB | | AWA | | FLO |
|---|---|---|---|---|---|
| | SS | PS | SS | PS | |
| LATEM (2016) | 49.4 | 49.3 | 74.8 | 55.1 | 40.4 |
| ALE (2015) | 53.2 | 54.9 | 78.6 | 59.9 | 48.5 |
| SJE (2015) | 55.3 | 53.9 | 76.7 | 65.6 | 53.4 |
| ESZSL (2015) | 55.1 | 53.9 | 74.7 | 58.2 | 51.0 |
| SYNC (2016) | 54.1 | 55.6 | 72.2 | 54.0 | - |
| SAE (2017) | 33.4 | 33.3 | 80.6 | 53.0 | 45.6 |
| DEM (2017) | 51.8 | 51.7 | 80.3 | 65.7 | 41.6 |
| GAZSL (2018) | 57.5 | 55.8 | 77.1 | 63.7 | 60.5 |
| SCoRe (2017) | 59.5 | 62.7 | 82.8 | 61.6 | 60.9 |
| LDF (2018) | 67.1 | 67.5 | 83.4 | 65.5 | - |
| Ours | **70.5** | **71.0** | **83.5** | **68.8** | **65.9** |

## 4.4 Zero-Shot Classification Results

We compare our method with two groups of state-of-the-art methods: non-end-to-end methods that use visual features extracted from pretrained CNN, and end-to-end methods that jointly train CNN and visual-sementic embedding network. The former group includes LATEM [9], ALE [8], SJE [42], ESZSL [23], SYNC [43], SAE [15], DEM [14], GAZSL [5], and the latter one includes SCoRe [13], LDF [19]. The evaluation results are shown in Table 2. Different groups of approaches are separated by a horizontal line. The scores of baselines (DAP-SAE) are obtained from [40, 10]. As the codes of DEM, GAZSL, SCoRe are available online, we obtain the results by running the codes on different settings if they are not published. We get all the results of LDF from the authors.

In general, we observe that the end-to-end methods outperform the non-end-to-end methods. That confirms that the joint training of the CNN model and the embedding model eliminates the discrepancy between features for conventional object recognition and those for zero-shot one that exists in non-end-to-end methods.

It's worth noting that LDF learns object localization by integrating an additional zoom network to the whole model, while our approach further involves part-level patches to provide local features of objects. It is clear that our proposed model consistently outperforms previous approaches, achieving impressive gains over the state-of-the-arts on fine-grained datasets: 3.4%, 3.5% on CUB SS/PS settings, and 5.0% on FLO. We find that the complexity of our model can be reduced by using the same CNN with shared weights for both image and part patches, but the zero-shot learning performance is slightly degraded, e.g., the score for CUB-PS and AWA-PS decreases by 3.6%, 2.7%, respectively.

We also evaluate our method on the generalized zero-shot learning setting, where the test images come from all classes including both seen and unseen categories. We report the performances of classifying test images from unseen classes and seen classes into the joint label space, denoted as $\mathcal{A}_\mathcal{U}$ and $\mathcal{A}_\mathcal{S}$ respectively, and the harmonic mean $\mathcal{H} = \frac{2 \cdot \mathcal{A}_\mathcal{S} \cdot \mathcal{A}_\mathcal{U}}{\mathcal{A}_\mathcal{S} + \mathcal{A}_\mathcal{U}}$.

Table 3: Generalized zero-shot learning results (%).

| Method | CUB | | | AwA | | |
|---|---|---|---|---|---|---|
| | $\mathcal{A}_\mathcal{U}$ | $\mathcal{A}_\mathcal{S}$ | $\mathcal{H}$ | $\mathcal{A}_\mathcal{U}$ | $\mathcal{A}_\mathcal{S}$ | $\mathcal{H}$ |
| DEM [14] | 19.6 | 57.9 | 29.2 | 32.8 | 84.7 | 47.3 |
| GAZSL [5] | 31.7 | 61.3 | 41.8 | 29.6 | 84.2 | 43.8 |
| LDF [19] | 26.4 | **81.6** | 39.9 | 9.8 | **87.4** | 17.6 |
| Ours | **36.7** | 71.3 | **48.5** | **37.6** | 87.1 | **52.5** |

As shown in Table 3, our model outperforms previous state-of-the-art methods with respect to $\mathcal{H}$ score. Especially in the CUB dataset where discriminative parts are crucial to capture the subtle difference among fine-grained classes, our method improves the $\mathcal{H}$ score by 6.7% (48.5% v.s. 41.8%).

## 4.5 Ablation Study

In this section, we study the effectiveness of the detected object regions and finer part patches, as well as the joint supervision of embedding softmax and class-center triplet loss. We set our baseline to be the model without localizing parts and with only embedding softmax loss as the objective.

**Effect of discriminative regions.** The upper part of Table 4 shows the performance of our method with different image inputs. Our model with only part regions performs worst because part regions only provide local features of an object, such as the features of head or leg. Although these local features are discriminative in the part level, it misses lots of information contained in other regions, and thus cannot recognize the whole object well alone. When we combine the original image and the localized parts, the performance has a significant improvement from the baseline by 5.4% (65.2% v.s. 59.8%).

To further demonstrate the effectiveness of the localized parts and objects, we combine the object with randomly cropped parts of the same part size. From the results, we observe, in most cases, adding random parts will hurt the performance. We believe it's due to the lack of alignment of random cropped parts. For instance, one random part in an image is roughly the head of the object, while it may focus on the leg in another image. In contrast, our localized parts have better semantic alignment, as shown in Figure 2.

Table 4: The performance of variants on zero-shot learning with PS setting. The best scores are marked bold.

| Method | CUB | AWA | FLO | Avg |
|---|---|---|---|---|
| Baseline | 60.2 | 61.5 | 57.7 | 59.8 |
| Parts | 55.4 | 51.2 | 49.8 | 52.1 |
| Baseline+Parts | **67.4** | **64.3** | **63.9** | **65.2** |
| Baseline+Random Parts | 56.3 | 59.8 | 56.4 | 57.5 |
| Embedding Softmax | 60.9 | 62.4 | 57.2 | 60.2 |
| Class-Center Triplet | 62.1 | 64.6 | 61.1 | 62.6 |
| Combined | **63.5** | **65.7** | **61.8** | **63.7** |

**Effect of joint loss.** The bottom part of Table 4 shows the results on different ways of inferences when our model is trained with the joint loss as the objective and only the original image as input. Compared with the baseline, the results inferred from the embedding softmax branch get improved a little as class-center triplet loss can be considered a regularizer to enhance the discriminative features. The results inferred from the class-center triplet branch are better, and we get the best results when combining the inferences of these two branches, which improves the baseline results by 3.9%.

## 5 Conclusion

In the paper, we show the significance of discriminative parts for zero-shot object recognition. It motivates us to design a semantic-guided attention localization model to detect such discriminative parts of objects guided by semantic representations. The multi-attention loss is proposed to favor compact and diverse attentions. Our model jointly learns global and local features from the original image and the discovered parts with embedding softmax loss and class-center triplet loss in an end-to-end fashion. Extensive experiments show that the proposed method outperforms the state-of-the-art methods.

**Acknowledgments**
This work is partially supported by NSFUSA award 1409683.

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
