[Supplementary Material · Appendix.pdf]

# Semantic-Guided Multi-Attention Localization for Zero-Shot Learning: Appendix

# Appendices

## A  Initialization of Attention Weights

In this section, we introduce how to initialize the attention weight $a$ in Eq. 2. Each channel of feature maps focuses on a certain pattern or a certain part of the object. We assume that if in several channels of features the positions of peak values are near to each other if not the same, these channels correspond to the same part of the object. Inspired by [1], we leverage the peak locations as the representations of feature channels and apply k-means clustering to cluster feature channels. Specifically, we feed all training samples into the CNN trained for the conventional classification task and extract the coordinates of the peak for each channel. Formally, the representation for $c^{th}$ channel is given by:

$$R_c = [x_1, y_1, x_2, y_2, ...x_n, y_n, ..., x_N, y_N], \tag{1}$$

where $x_n, y_n$ are the coordinates of the peak for $n^{th}$ sample, $N$ is the number of samples. Applying k-means clustering, we obtain the n=2 groups of channels corresponding to two attention maps. We initialize the channel-wise attention weights for $i^{th}$ part by an indicator function over all feature channels:

$$\hat{a}_i = [\mathbb{1}(1), \mathbb{1}(2), ..., \mathbb{1}(c), ..., \mathbb{1}(C)], \tag{2}$$

where $\mathbb{1}(\cdot)$ equals one if the $c^{th}$ channel belongs to the $i^{th}$ cluster and zero otherwise, and $C$ is the number of channels. With these pseudo values of $a_i$, we can initialize the stacked FC layers in Eq. 2 by the regression with $L_2$ loss:

$$\min_{W_1, W_2} ||\sigma(W_2 f(W_1 p)) - \hat{a}_i||_2^2 \tag{3}$$

The stacked FC layers, as a part of our model, are optimized later with the SGMA objective in an end-to-end fashion.

## B  Initialization of Cropping Subnet

We leverage the attention maps from the initialized multi-attention subnet to pretrain the region cropping subnet. Specifically, we obtain the attended region in attention maps by a discriminative square centered at the peak response of the attention map. The side length of the squares $t_s$ is assumed to be the quarter of the image size. The coordinates of the attended region ($[t_x, t_y, t_s]$) are utilized to pretrain the cropping subnet with $L_2$ loss.

## C  Implementation details

We implement our approach on the Pytorch Framework. For the multi-attention subnet, we take the images of size 448 by 448 as input to achieve high-resolution attention maps. For joint feature

embedding subnet, we resize all the input images to the size 224 by 224. We consistently adopt VGG19 as the backbone and train the model with the batch size of 32 on two GPUs(TitanX). We use SGD optimizer with the learning rate of $0.05$, the moment of $0.9$ and weight decay of $5 * 10^{-4}$ to optimize the objectives. The learning rate is decay by $0.1$ on the plateau, and the minimum one is set to $5 * 10^{-4}$. Hyper-parameters in our models are obtained by grid search on the validation set. $mrg$s in Eq. 6 and Eq. 11 are set to 0.2 and 0.8 respectively. $k$ in Eq. 8 is set to 10.

## D    Inference from SGMA model

We provide two ways to infer the labels of unseen class images from the SGMA model. The first one is straightforwardly to choose the class label with the maximal overall compatibility score, as the green path in Figure 2. An alternative way is utilizing the features $\phi_{cct}(x)$ learned in the class-center branch, as the purple path in Figure 2. We employ the inner product to measure the similarities between the feature of the test image $\phi_{cct}(x)$ and the prototypes of unseen classes $\Phi^u_{cct}$, which can be obtained by the following steps. The prototypes of seen classes $\Phi^s_{cct}$ is obtained by averaging the features of all images of each class. We assume the semantic descriptions of unseen classes can be represented by a linear combination of those of seen classes. Let $W$ be the weight matrix of such a combination, and $W$ can be obtained by solving the ridge regression:

$$W = \arg \min_{W} ||\Phi^u - W\Phi^s||_2^2 + \lambda ||W||_2^2, \tag{4}$$

where $\Phi^u$ and $\Phi^s$ are the semantic matrices of unseen and seen classes with each row being the semantic vector of each class. Equipped with the learned $W$ describing the relationship of the semantic vectors of seen and unseen classes, we can obtain the prototypes for unseen classes by applying the same $W$, $\Phi^u_{cct} = W\Phi^s_{cct}$.

To combine the global and local descriptions of images, we concatenate the visual features generated by different CNNs. Moreover, to combine the inference of two ways, the compatibility scores of the embedding softmax branch and the inner product of the class-agent triplet branch are added as the final prediction scores of the test image w.r.t. unseen classes:

$$y = \arg \min_{y \in \mathcal{Y}_{\mathcal{U}}} (s_y + \langle \phi_{cct}(x), [\Phi^u_{cct}]_y \rangle), \tag{5}$$

where $s_y = \theta(x)^T W \phi(y)$, and $[\cdot]_y$ denotes the row of the matrix corresponding to the class $y$.

## E    Evaluation on Different Semantic Embedding

In the section, we evaluate the performance of our model with different semantic embedding. The alternative semantic information we are using is Wikipedia articles, word2vec and fasttext embedding of class names. The work [2] collected Wikipedia articles associated with each class. They extracted the TF-IDF feature as semantic representations of classes. We observe that the high-dimensional TF-IDF features ($\sim$7000D) are sparse, so PCA is employed to compress them to 200D feature for efficiency. We directly use the 500D word2vec feature provided by [3], and extract the 300D fasttext feature from the model trained on Wikipedia-2017. Figure 1 shows the comparison of our SGMA model with four state-of-the-art methods. Overall, our SGMA model outperforms others in all cases. It is not surprising that attributes enable the most effective transfer as attributes are defined to be discriminant properties of object classes. Although Wikipedia articles are highly suffering from noise, they are still more informative and discriminative semantic descriptors than word embeddings. Comparing the results of two kinds of word embedding methods (i.e., word2vec, fasttext), we can find little difference.

## F    t-SNE Visualizations of Learned Features

We visualize the image features for each class in AwA1 dataset using t-SNE [4] as shown in Figure 2. It is obvious that our trained features are more discriminative cross the classes compared with the feature without finetuning. Another notable observation is that the features from the triplet branch are more intra-class compact and inter-class separatable, explaining the better performance as shown in Table 3.

Figure 1: Zero-shot learning results on CUB with various types of semantic embeddings.

Figure 2: t-SNE visualization of feautures. (a) the features extracted from resnet101 pretrained on Imagenet, (b) the features from embedding softmax embedding branch, (c) the features from class-agent triplet branch.

## G   More Quantitive Results for Part Detection

Here we provide more qualitative results to demonstrate the capability of our model in weakly-supervised part detection. Results on CUB, AwA2, FLO are shown in Figure 3a& 3b& 3c, respectively.

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

(a) **CUB**
(b) **AWA2**
(c) **FLO**

Figure 3: Part detection results