[Reviews · NeurIPS 2019]

Reviewer 1



The problem is relevant and the method is based on an interesting attention based idea to look at different regions in the image for the task of ZSL The losses used focus on (i) making each attention map peaky, while making different maps diverse, (ii) embedding based softmax for better prediction and (iii) class center triplet loss which makes the features closer to their respective class centers relative to the other class centers. Line 190 mentions that the image and parts are sent to “separate backbone networks”, which implies that the network parameters are not shared. If that is the case then the method will have ~3x parameters cf competing methods ie. a significantly higher capacity network overall. What happens when the CNN params are shared? And what happens when the image only baseline has a higher capacity network backbone (which is also then end-to-end finetuned)? The learning of channel wise attention weights are initialized by clustering the features using k-means, which is shown as an L2 loss minimization approach in eqn.(3) in supplementary section. A detailed ablation study would have been helpful showing the importance of this initialization. The number of clusters is fixed to be 2 for all the datasets. There is no justification or experiments provided to validate the same. Parametric studies for this should be provided, preferably with network weight sharing and without. As mentioned in supplementary material, clustering of feature channels is done using “CNN trained for the conventional classification task and extract the coordinates of the peak for each channel”. Is this a diffrent CNN other than that used in the approach? If so then the approach should include the same or else if the clustering is done along with the training of the CNN network embedded in the approach, then it would give erroneous peak channel value for the initial iterations. A more clear explanation is required about the calculation of channel wise attention. Results for Generalized ZSL, which is a more practical and harder task combining both seen and unseen classes at test time. Missed one of the relevant approaches in this area: Ji, Zhong, et al. "Stacked semantics-guided attention model for fine-grained zero-shot learning." Advances in Neural Information Processing Systems. 2018. A more recent paper in the same idea of attention for ZSL (although the paper appeared after NeurIPS’19 submission date, so not considering in this review, only fyi): Xie, Guo-Sen, et al. "Attentive Region Embedding Network for Zero-shot Learning." Proceedings of the IEEE Conference on Computer Vision and Pattern Recognition. 2019. Minor comments: ‘qualitative’ in place of ‘quantitative’ as mentioned in the title of appendix G Some of the main contributions are pushed to supplementary section, such as the details of clustering of channels mentioned in appendix A and the inference in appendix D. It would have been better if these contents could have been mentioned in the main paper. --------------------- Post rebuttal --------------------- I appreciate the interesting rebuttal. However, I still find that the submission would need more work. - The with and without parameter sharing exposes high variance in results with number of clusters. Since the part definition step is stochastic, I would also do multiple initializations and report variances. - The ZSL results are not quite state of the art (eg many results from CVPR 2018 Feature Generating networks, Xian et al.; there are even better numbers out there now) - The clustering process and its working would still need more analysis and discussion (with experiments) I would keep my rating.

Reviewer 2



While there are weaknesses, this paper is a solid submission. The idea is interesting and effective. It outperforms the state of the art. Strength: + The paper is well written and the explanations are clear. + The quantitative results (especially Table 2) clearly demonstrate the effectiveness of the proposed method. + Figure 1 is well designed and useful to understand the model. + Qualitative results in Figure 2 is convincing and demonstrates the consistency of the attention module across different classes. Weakness: - Motivation behind 3.2 Section 3.2 describes the cropping network that uses a 2d continuous boxcar function. Motivation for this design choice is weak, as previous attempts in local attention have used Gaussian masks [a], simple bilinear sampling using spatial transformers [b], or even pooling methods [c]. If this makes a difference, it would be great to demonstate it in an experiment. At minimum, bilinear sampling should be compared against. [a] Gregor, Karol, et al. "Draw: A recurrent neural network for image generation." ICML, 2015. [b] Jaderberg, Max, Karen Simonyan, and Andrew Zisserman. "Spatial transformer networks." NeurIPS, 2015. [c] He, Kaiming, et al. "Mask r-cnn." Proceedings of the IEEE international conference on computer vision. 2017. - Discrepancy between eq. 9 and Figure 1. From eq. 9, it seems like the output patches are not cropped parts of the input image but just masked versions of the input image where most pixels are black. Is this correct? In this case, Figure 1 is misleading. And if so, wouldn't zooming on the region of interest using bilinear sampling provide better results? - Class-Center Triplet Loss The formulation of class-center triplet loss (L_CCT) is not entirely convincing. While the authors claim L2 normalization is introduced to ease the setting of a proper margin, this also has a different effect. This would in fact, divert the formulation to be different from the traditional definition of a margin. For example, these two points in the semantic feature space could be close, but far away after the normalization that projects them on a unit hypersphere. And the other way around is also true. Especially given the fact that the unnormalized version of phi is used also in L_CLS, the effect of this formulation is not obvious. In fact, the formulation resembles the cosine distance in an inner product, and the margin would be set -- roughly speaking -- on the cosine angle. The authors should discuss this in their paper. I find the current explanation misleading. - Backbone CNN Although I assume so, in Section 3.3 / Figure 1, it is not clear which backbone CNNs share their weights, and which don't (if some don't). Is the input image going through the same CNN as the local patches? Are the local patches going through the same CNN? I suggest some coloring to make it clear if not all are shared. - Minor issues L15: "must be limited to one paragraph". L193: L_CAT --> L_CCT Equation 11: it would be clearer with indices under the max function. L215: "unit sphere" -> "unit hypersphere". Unless the dimension of the semantic feature space is 3, which in this case should be mentioned. Potential Enhancements: * This paper is targeting zero-shot classification but since the multi-attention module is a major contribution by itself, it could have been validated on other tasks. An obvious one is fine-grained classification, on CUB-200 for instance. It is maybe possible for the authors to report this result since they already use CUB-200, but I would understand if it is not done in the rebuttal. ==== POST REBUTTAL ==== The additional results have made the submission even stronger than before. I am therefore more confident in the rating.

Reviewer 3



Originality: The work combines a set of well known components from the DL community. We do not see a real technical innovation, but the combination of the said methods is interesting in itself. The difference o the Multi-Attention compared to previous methods, is not clearly explained and not very convincing. Clarity: I find that the submission is clear overall. About the class-center triplet loss; The authors should explain how the extracted features are finally used for 0-shot learning. This gets clearer in the supplementary material, but it should be explained in the main paper and a reference to the supplementary should be added. A couple of typos - one in the abstract-. Figure 1. and the text of the paper clearly describe how the multiple components are articulated. Significance: The impact and significance of the paper for the community will be average. This is a good piece of work and the code is provided, nevertheless, we do not see a clear technical breakthrough that would be reused by other authors.

[Author Response · NeurIPS 2019]

| Model | # of params | #of clusters | CUB-SS | CUB-PS | AwA-SS | AwA1-PS | FLO | Average |
|---|---|---|---|---|---|---|---|---|
| LDF(2018) [1] | 426.4M | - | 67.1 | 67.5 | 83.4 | 65.5 | - | - |
| Resnet152 | 60.2M | - | 66.9 | 67.3 | 81.0 | 67.5 | 64.0 | 69.3 |
| Ours w/ SPN | 61.0M | 2 | 70.1 | 70.5 | **83.7** | 68.5 | 64.2 | 71.4 |
| Ours | 61.0M/42.5M | 2 | **70.5**/66.5 | 71.0/67.4 | 83.5/**82.9** | 68.8/66.1 | **65.9/65.6** | **71.8/69.7** |
|  | - | 3 | 69.2/**67.3** | **71.7**/67.1 | 82.4/82.6 | 66.3/66.5 | 65.8/64.7 | 71.1/69.4 |
|  | - | 4 | 70.2/67.1 | 71.3/**67.6** | 82.0/81.9 | 68.4/65.9 | 64.2/**65.6** | 71.2/69.6 |

Table 1: Zero-shot learning results on three benchmarks. The number of parameters is calculated for the CUB dataset. We report the results of our model **without/with** sharing the CNN parameters for the input image and the local patches.

| Method | CUB $A_{\mathcal{U}\to\mathcal{T}}$ | CUB $A_{\mathcal{S}\to\mathcal{T}}$ | CUB $H$ | AwA1 $A_{\mathcal{U}\to\mathcal{T}}$ | AwA1 $A_{\mathcal{S}\to\mathcal{T}}$ | AwA1 $H$ | AwA2 $A_{\mathcal{U}\to\mathcal{T}}$ | AwA2 $A_{\mathcal{S}\to\mathcal{T}}$ | AwA2 $H$ | SUN $A_{\mathcal{U}\to\mathcal{T}}$ | SUN $A_{\mathcal{S}\to\mathcal{T}}$ | SUN $H$ |
|---|---|---|---|---|---|---|---|---|---|---|---|---|
| DEM [2] | 19.6 | 57.9 | 29.2 | 32.8 | 84.7 | 47.3 | 30.5 | 86.4 | 45.1 | 20.5 | 34.3 | 25.6 |
| RN* [3] | **38.1** | 61.1 | 47.0 | 31.4 | **91.3** | 46.7 | 30.0 | **93.4** | 45.3 | 20.1 | 35.6 | 25.7 |
| LDF* [1] | 26.4 | **81.6** | 39.9 | 9.8 | 87.4 | 17.6 | - | - | - | - | - | - |
| Ours* | 36.7 | 71.3 | **48.5** | **37.6** | 87.1 | **52.5** | **36.0** | 84.3 | **50.5** | **22.3** | **39.5** | **28.5** |

Table 2: Generalized zero-shot learning results (%). $H$ denotes the harmonic mean. * means end-to-end training.

Figure 1: Training curve.

We first thank all reviewers for the valuable feedback.

**Reviewer1**

**Q1**: **Much more parameters.** To prove that the gain of performance is not totally from the higher capacity network, we conduct experiments using Resnet152 as the backbone with end-to-end finetune, which has a comparable amount of parameters to ours. As shown in Table 1, our model outperforms Resnet152 by 3.6%(71.8% v.s. 69.3%). Besides, our model has **significantly less parameters** than the best competing model LDF [1] while performing much better. We also can reduce the number of parameters by using the same CNN for the image and the part patches as you suggested, but the ZSL performance is slightly degraded roughly from 71% to 69% as shown in Table 1 (separated with slashes).

**Q2: The importance of weights initialization.** We present the training curves of our model with/without weights initialization in Fig 1. We see that initializing the attention layers speeds up the learning and finally achieves a greater accuracy. We will add more detailed analysis in the final version of the paper.

**Q3: The number of clusters.** As shown in Table 1, we increase the number of clusters to 4 and find little performance improvement. Besides, we observe more maps introduce the attention redundancy, i.e. maps attend to the same region.

**Q4: Results for generalized ZSL**. As shown in Table 2, our model outperforms the other SOTA models (based on $H$).

**Other comments**. The CNN pretrained to provide psuedo labels for clustering is the same backbone used in our model, otherwise it would give erroneous peak as you agreed. We will cite the relevant papers you suggest.

**Reviewer2**

**Q1: About Cropping network.** In fact, to obtain better representation for finer localized cropped region $x_i^{part}$, our method also utilizes the bilinear sampling to adaptively zoom the cropped region $x_i^{part}$ to the same size with the original image. Concretely, for a point (i, j) of the zoomed region, its value $x_{(i,j)}^{zoom}$ can be computed bilinearly combining the values of nearest four points in the cropped region. Formally, $x_{(i,j)}^{zoom} = \sum_{\alpha,\beta} |1 - \alpha - \{i/\lambda\}||1 - \beta - \{j/\lambda\}| x_{(m,n)}^{part}$, where $m = [i/\lambda] + \alpha + z_x - z_s, n = [j/\lambda] + \beta + z_y - z_s$, $\alpha = 0$ or 1, $\beta = 0$ or 1, $\lambda$ is the upsampling factor, $\lambda = t/t_s$ ($t$ is the size of the original image) and $[\cdot]$ and $\{\cdot\}$ is the integral and fractional part, respectively. We will add the detailed description in the final version of the paper. Spatial Transformer Network(SPN) is an alternative of our cropping net. When replacing it with SPN, we find the performance changes little as shown in Table 1.

**Q2: About triplet loss.** We agree that the normalization will change the relative distance of two points. There is a typo leading to misunderstanding in the paper. We actually use the normalized version of $\phi$ in the embedding softmax loss so that only normalized features are considered and used in training and inference phases. Please refer to [4]. We will add more discussion in the final version of paper. **Other comments:** The local patches are going through the same CNN while the input image is going through a different CNN. We will mark it in the figure.

**Reviewer3**

The competing method LDF [1] used a single attention scheme and we have shown our superiority to it in both the model design and the performance. We will add more explanation about how the extracted features are used and add a reference to the appendix. Please kindly refer to our response to other reviewers.

# References

[1] Li et al. Discriminative learning of latent features for zero-shot recognition. In *CVPR*, pages 7463–7471, 2018.

[2] Zhang et al. Learning a deep embedding model for zero-shot learning. In *CVPR*, pages 2021–2030, 2017.

[3] Sung et al. Learning to compare: Relation network for few-shot learning. In *CVPR*, pages 1199–1208, 2018.

[4] Wang et al. Normface: l2 hypersphere embedding for face verification. In *ACMMM*, pages 1041–1049. ACM, 2017.


[Meta-Review · NeurIPS 2019]

The contribution is interesting and proposes novel ideas for zero-shot learning based on attention mechanisms and combining different local and global features. The method achieves good results, the rebuttal helps to strengthen the contribution.